# Long-term survival outcomes of endoscopic therapy vs. surgical resection in patients with cardia gastrointestinal stromal tumor

Qiong Wu[1☯], Jun Jiang[2☯], Zhuofan Li[3☯], Xin Ling[1], Zhenguo Qiao[1]*, Yimin Ma[4]*

**1** Department of Gastroenterology, Suzhou Ninth People's Hospital, Suzhou Ninth Hospital Affiliated to Soochow University, Suzhou, China, **2** Department of Public Health and Preventive Medicine, Anhui Medical College, Hefei, China, **3** Suzhou Medical College of Soochow University, Suzhou, China, **4** Departments of Gastroenterology, Gaochun People's Hospital of Nanjing, Nanjing, China

☯ These authors contributed equally to this work.

* qzg66666666@163.com (ZQ); kelema917@sina.com (YM)

**Data Availability Statement:** Publicly available datasets were analyzed in this study. These data can be found here: https://seer.cancer.gov/. The

## Abstract

The ideal surgical approach for treating cardia gastrointestinal stromal tumor (GIST) is not clearly established. This study aimed to assess the long-term survival results among patients who received endoscopic therapy (ET) or surgical resection (SR) for cardia GIST. Cardia GIST patients from 2000 to 2019 were selected from the surveillance, epidemiology, and end result (SEER) database. Multiple imputation (MI) was applied to handle missing data, and propensity score matching (PSM) was carried out to mitigate selection bias during comparisons. Demographic and clinical characteristics' effects on overall survival (OS) and cancer-specific survival (CSS) were assessed using Kaplan-Meier analyses and multivariate Cox proportional hazard models. A total of 330 patients with cardia GIST were enrolled, including 47 (14.2%) patients with ET and 283 (85.8%) patients with SR. The 5-year OS and CSS rates in the ET and SR groups were comparable [before PSM, (OS) (76.1% vs. 81.2%, $P = 0.722$), (CSS) (95.0% vs. 89.3%, $P = 0.186$); after PSM, (OS) (75.4% vs. 85.4%, $P = 0.540$), (CSS) (94.9% vs. 92.0%, $P = 0.099$)]. Moreover, there was no significant difference between ET and SR in terms of long-term OS (hazard ratio [HR] 0.735, 95% confidence interval [CI] 0.422–1.282) and CSS (HR 1.560, 95% CI 0.543–4.481). Our study found no significant disparity in long-term survival outcomes between ET and SR in cardia GIST patients, implying that ET could be a valid surgical strategy for treating cardia GIST.

## Introduction

Gastrointestinal stromal tumor (GIST) is a type of cancer that originates in the gastrointestinal tract, typically in the stomach or small intestine [1,2]. Cardia GIST is rare, accounting for about 8.7% to 17.0% of all GIST [3–5]. Although surgical resection (SR) has been the conventional treatment method for GIST, the optimal surgical strategy for cardia GIST, a tumor situated in the upper region of the stomach, is uncertain [6]. With the recent rapid advancement of endoscopic technology, endoscopic therapy (ET) has emerged as a viable and less invasive

datasets supporting the conclusions of this article are included within the article.

**Funding:** The author(s) received no specific funding for this work.

**Competing interests:** The authors have declared that no competing interests exist.

alternative to SR for managing GIST. ET offers advantages such as reduced trauma, less intraoperative bleeding, and a lower occurrence of complications as compared to SR [7,8].

However, most of the recent studies on ET for treating GIST have been centered on the stomach, and there is still insufficient data available regarding the application of ET for cardia GIST [9,10]. Meanwhile, the impact of ET and SR on the long-term survival of patients diagnosed with cardia GIST remains uncertain. Therefore, this study aims to conduct a comparison between ET and SR in terms of long-term survival outcomes for patients diagnosed with cardia GIST, in order to provide valuable insights that can assist clinicians and patients in making informed decisions regarding the choice of treatment.

## Methods

### Patient selection

Patients with GIST who were diagnosed between 2000 and 2019 were selected from the SEER database (http://seer.cancer.gov). GIST identification and primary tumor site were based on the third edition of the International Classification of Diseases for Oncology (ICD-O-3) with a specific histologic subtype code of 8936 and a primary site code of C16.0. Exclusion criteria included: (1) no primary site surgery; (2) gastrectomy with a resection of other organs; (3) unknown surgery information; (4) unknown survival information. Age, sex, race, grade, tumor size, chemotherapy, mitotic count, follow-up time, cancer-specific death, and vital status were among the extracted covariates. The RX Summ-Surg Prim Site (1998+) codes were utilized to differentiate between various surgical techniques, such as ET (codes 20, 26, 27) and SR (codes 30–52). The purpose of the study was to analyze the overall survival (OS) and cancer-specific survival (CSS) of patients with cardia GIST who underwent either ET or SR. OS and CSS were determined from the time of diagnosis of cardia GIST to the date of death or death from cancer or the most recent follow-up.

### Ethics statement

Data from the SEER database were used to extract information for institutional cohorts. The authors did not conduct any studies involving human participants, so formal consent was not necessary for this type of research.

### Statistical analysis

Categorical variables were presented as frequencies and percentages, and statistical comparisons between groups were performed using either the Chi-square test or Fisher exact test. To handle missing data, a polytomous regression model was developed based on patient race, grade, tumor size, marital status, and mitotic count using multiple imputation (MI). This procedure was repeated five times to produce a final dataset. One to one propensity score matching (PSM) was used to match ET and SR groups, with the propensity model incorporating age, race, sex, grade, tumor size, marital status, chemotherapy, and mitotic count, and the caliper width set to 0.02. The Kaplan-Meier method was employed to calculate overall survival (OS) and cancer-specific survival (CSS), and the log-rank test was used to compare them. Multivariate Cox proportional hazard models were used to assess hazard ratios (HRs) and 95% confidence intervals (CIs). SPSS version 26 (Chicago, IL, USA) was used for all statistical analyses, while Kaplan-Meier curves were created and log-rank tests were performed using R software (version 4.0). A significance level of $P < 0.05$ was set to identify significant differences between groups.

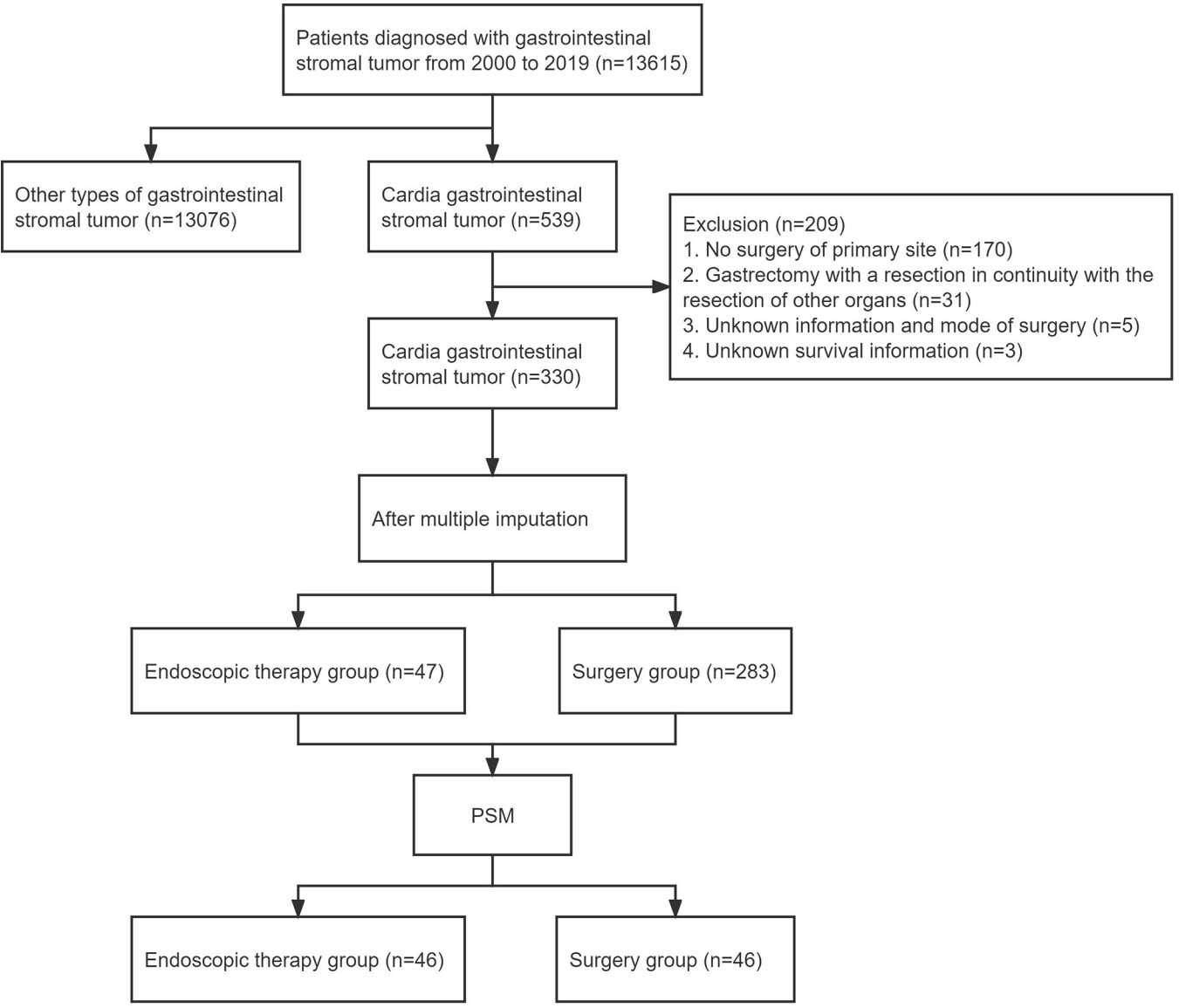

**Fig 1. Flow chart of the study.** PSM: Propensity score matching.

## Results

### Patients characteristics

The process of selecting patients in this study is presented in Fig 1. A total of 330 patients with cardia GIST who met the inclusion criteria were enrolled in the study, of whom 47 (14.2%) underwent ET and 283 (85.8%) underwent SR. The patients' baseline characteristics are summarized in Table 1.

### Comparison of ET and SR outcomes

Before PSM, there was a statistically significant difference ($P < 0.05$) in tumor size between the ET and SR groups. The median follow-up period for the ET and SR groups was 66 (range 0–230) months and 69 (range 0–238) months, respectively. Comparable OS and DSS outcomes were observed in the ET and SR groups. The 5-year OS rate was 76.1% in the ET group and

**Table 1. Baseline characteristics of patients before and after PSM.**

| | Before PSM | | | After PSM | | |
|---|---|---|---|---|---|---|
| | ET group (n = 47) | Surgery group (n = 283) | *P*-value | ET group (n = 46) | Surgery group (n = 46) | *P*-value |
| Age, year, n (%) | | | 0.791 | | | 0.514 |
| ≤ 60 | 16 (34.0) | 102 (36.0) | | 15 (32.6) | 18 (39.1) | |
| > 60 | 31 (66.0) | 181 (64.0) | | 31 (67.4) | 28 (60.9) | |
| Race, n(%) | | | 0.561 | | | 0.521 |
| White | 32 (68.1) | 170 (60.1) | | 31 (67.4) | 26 (56.5) | |
| Black | 8 (17.0) | 56 (19.8) | | 8 (17.4) | 12 (26.1) | |
| Other | 7 (14.9) | 57 (20.1) | | 7 (15.2) | 8 (17.4) | |
| Sex, n(%) | | | 0.909 | | | 0.532 |
| Male | 25 (53.2) | 148 (52.3) | | 24 (52.2) | 21 (45.7) | |
| Female | 22 (46.8) | 135 (47.7) | | 22 (47.8) | 25 (54.3) | |
| Grade, n (%) | | | 0.460 | | | 0.198 |
| Well/ Moderately | 40 (85.1) | 228 (80.6) | | 39 (84.8) | 34 (73.9) | |
| Poorly/ Undifferentiated | 7 (14.9) | 55 (19.4) | | 7 (15.2) | 12 (26.1) | |
| Tumor size, cm, n (%) | | | **0.008** | | | 0.826 |
| ≤ 2.0 | 12 (25.5) | 26 (9.2) | | 11 (23.9) | 10 (21.7) | |
| 2.0–5.0 | 18 (38.3) | 108 (38.2) | | 18 (39.1) | 19 (41.3) | |
| 5.0–10.0 | 13 (27.7) | 123 (43.5) | | 13 (28.3) | 15 (32.6) | |
| > 10.0 | 4 (8.5) | 26 (9.2) | | 4 (8.7) | 2 (4.3) | |
| Marital status, n (%) | | | 0.603 | | | 0.815 |
| Married | 34 (72.3) | 194 (68.6) | | 33 (71.7) | 34 (73.9) | |
| Unmarried | 13 (27.7) | 89 (31.4) | | 13 (28.3) | 12 (26.1) | |
| Chemotherapy, n (%) | | | 0.075 | | | 0.650 |
| No | 34 (72.3) | 166 (58.7) | | 33 (71.7) | 31 (67.4) | |
| Yes | 13 (27.7) | 117 (41.3) | | 13 (28.3) | 15 (32.6) | |
| Mitotic count, HPF, n (%) | | | 0.398 | | | 0.189 |
| ≤ 5/50 | 34 (72.3) | 187 (66.1) | | 33 (71.7) | 27 (58.7) | |
| > 5/50 | 13 (27.7) | 96 (33.9) | | 13 (28.3) | 19 (41.3) | |

PSM: Propensity score matching; ET: Endoscopic therapy; Other: American Indian, Alaska Native, Asian/Pacifc Islander; HPF: High power field.

81.1% in the SR group (*P* = 0.722, Fig 2A), whereas the 5-year CSS rate was 95.0% in the ET group and 88.7% in the SR group (*P* = 0.186, Fig 2C). After PSM, the analysis included 46 matched pairs of patients, with no significant difference observed in any baseline characteristics between the two groups (Table 1). The median follow-up period was 65 (range 0–230) months for the ET group and 71 (range 0–196) months for the SR group. The survival outcomes of the patients in the ET group were comparable to those in the SR group. The 5-year OS rate was 75.4% in the ET group and 85.4% in the SR group (*P* = 0.540, Fig 2B), whereas the 5-year CSS rate was 94.9% in the ET group and 92.0% in the SR group (*P* = 0.099, Fig 2D).

## Prediction results of the multivariate COX model

The multivariate Cox analysis results are shown in Table 2. The Cox proportional hazards regression analysis indicated that the type of surgical strategy had no significant effect on OS (hazard ratio [HR] 0.735, 95% confidence interval [CI] 0.422–1.282, *P* = 0.279) and CSS (HR 1.560, 95% CI 0.543–4.481, *P* = 0.409). Additionally, our study found that age and sex were significant prognostic factors for OS, while poorly and undifferentiated grade was a significant prognostic factor for CSS (Table 2).

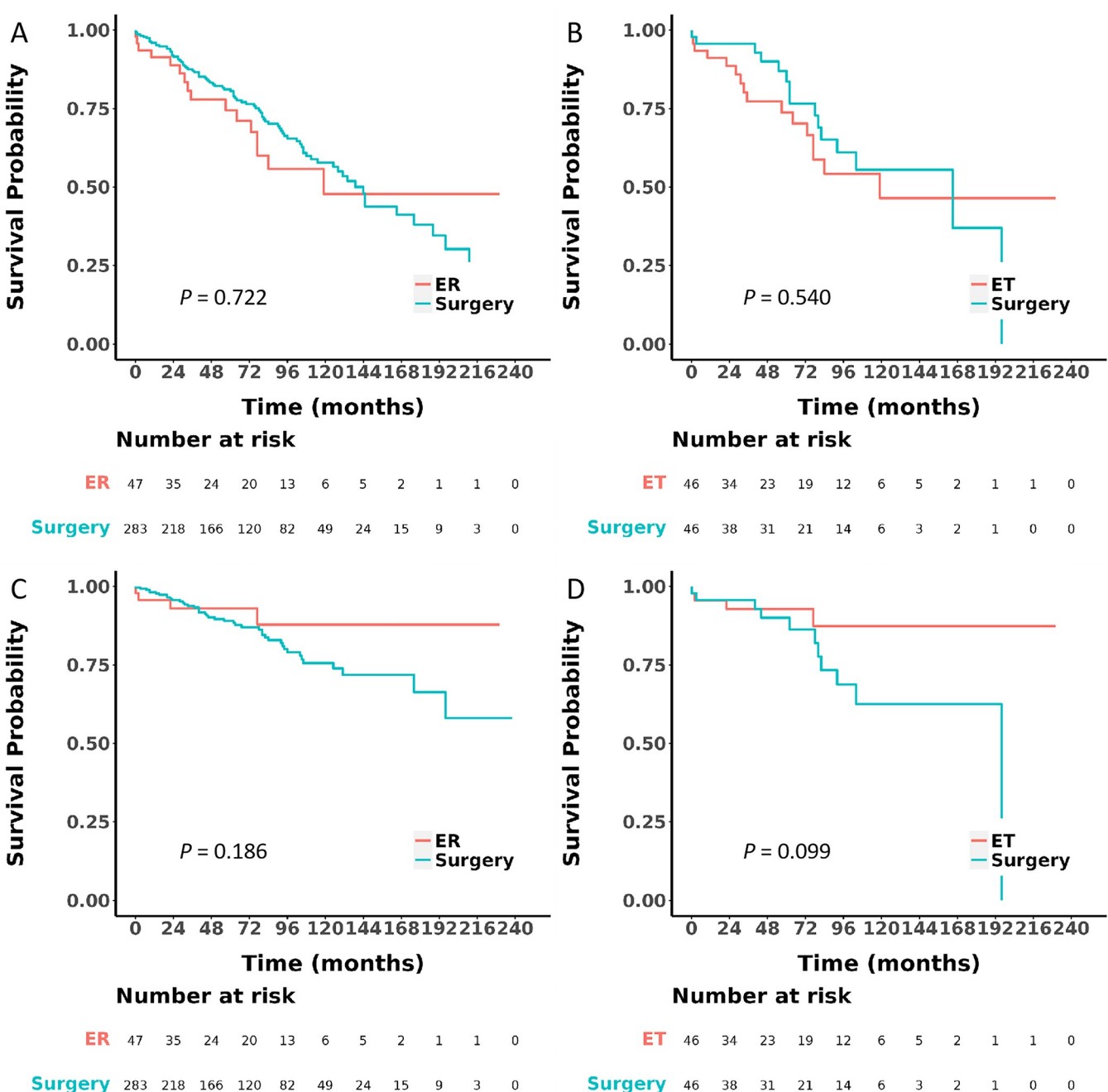

**Fig 2.** Overall survival and cancer-specific survival were compared before and after PSM (A and C before PSM, and B and D after PSM) after multiple imputation was performed.

## Discussion

Surgical resection is currently the preferred treatment method for GIST [11,12]. The primary guidelines for the surgical procedure involve achieving tumor-free resection margins through complete excision, while minimizing any compromise to organ functions [13]. However, the occurrence of GIST in the cardia region is infrequent. Resection of these tumors is often

**Table 2. Multivariate analysis on overall survival and cancer-specific survival.**

| | OS | | CSS | |
|---|---|---|---|---|
| | HR(95%CI) | *P*-value | HR(95%CI) | *P*-value |
| Age, years, n(%) | | | | |
| ≤ 60 | Reference | | Reference | 0.159 |
| > 60 | 2.938(1.803–4.785) | **<0.001** | 1.589(0.834–3.025) | |
| Race, n(%) | | | | |
| White | Reference | | Reference | 0.702 |
| Black | 0.963(0.557–1.666) | 0.894 | 1.173(0.518–2.660) | 0.217 |
| Other | 1.081(0.624–1.874) | 0.780 | 1.609(0.756–3.422) | |
| Sex, n(%) | | | | |
| Male | Reference | | Reference | 0.219 |
| Female | 0.580(0.374–0.899) | **0.015** | 0.670(0.354–1.268) | |
| Grade, n (%) | | | | |
| Well/ Moderately | Reference | | Reference | **0.017** |
| Poorly/Undifferentiated | 1.430(0.757–2.701) | 0.270 | 3.052(1.225–7.603) | |
| Tumor size, cm, n (%) | | | | |
| ≤ 2.0 | Reference | | Reference | 0.894 |
| 2.0–5.0 | 1.176(0.559–2.474) | 0.669 | 0.911(0.233–3.570) | 0.720 |
| 5.0–10.0 | 1.253(0.596–2.637) | 0.551 | 1.275(0.338–4.810) | 0.209 |
| > 10.0 | 2.277(0.945–5.485) | 0.067 | 2.528(0.594–10.753) | |
| Marital status, n (%) | | | | |
| Married | Reference | | Reference | 0.521 |
| Unmarried | 1.077(0.689–1.686) | 0.744 | 1.228(0.656–2.302) | |
| Chemotherapy, n (%) | | | | |
| No | Reference | | Reference | 0.717 |
| Yes | 0.755(0.468–1.220) | 0.251 | 1.130(0.583–2.191) | |
| Mitotic count, HPF, n (%) | | | | |
| ≤ 5/50 | Reference | | Reference | 0.400 |
| > 5/50 | 0.910(0.518–1.600) | 0.744 | 0.682(0.280–1.663) | |
| Treatment, n (%) | | | | |
| ET | Reference | | Reference | 0.409 |
| Surgery | 0.735(0.422–1.282) | 0.279 | 1.560(0.543–4.481) | |

MI: Multiple imputation; OS: Overall survival; CSS: Cancer-specific survival; Other: American Indian, Alaska Native, Asian/Pacifc Islander; HPF: High power field; ET: Endoscopic therapy.

difficult due to their anatomical location, which may require proximal gastrectomy, and can result in severe postoperative gastroesophageal reflux and poor quality of life [14]. With the advancement and widespread use of endoscopic techniques, an increasing number of endoscopists are treating GIST located in the cardia region using endoscopic methods [15,16]. However, due to the relatively low incidence of cardia GIST, there is currently no research evaluating the long-term effects of ET or SR on the survival of these patients. To our knowledge, this is the first study to investigate the long-term effects of ET and SR on the survival of patients with cardia GIST.

Our results showed that the 5-year OS and 5-year CSS were not significantly different between the groups that underwent ET and those that received SR, and the Cox proportional hazards regression analysis indicated that the type of surgical approach did not significantly impact the patients' prognosis. The similar long-term prognosis of GIST patients undergoing

different surgical approaches may be attributed to several factors: (1) The targeted drug imatinib can effectively control tumor growth and spread, especially for those with high-risk GIST according to postoperative pathological evaluation, thus prolonging patients' survival [17,18]; (2) GIST is typically a solitary tumor with a clear peseudocapsule, resulting in a lower potential for invasion and metastasis compared to other malignant tumors, which improves the patient's prognosis to some extent; (3) With the development of medical technology, most GIST cases can be diagnosed early, and both ET and SR can effectively treat early-stage GIST, leading to better patient prognosis; (4) Complete tumor resection can lead to a cure as GIST rarely metastasizes to lymph nodes, and both ET and SR can completely remove the tumor.

Kramer et al.'s [19] study demonstrated that younger age and female gender were significantly associated with a more favourable prognosis in GIST, which is consistent with our study findings. Meanwhile, Tran et al [20] reported that older age (>65 years) is an independent predictor for mortality (OS) in GIST patients. Additionally, previous studies have consistently shown that female patients have a better long-term prognosis than male patients [21–23]. Khan et al. [24] and Güller et al. [25] reported similar findings to our study, demonstrating that male gender and age > 60 years were linked to poorer overall survival. Older GIST patients have a poorer prognosis, possibly because they may have comorbidities and weakened immune systems, making them more vulnerable to the negative effects of cancer and cancer treatments. The reasons why female GIST patients have a better prognosis than male patients are not entirely clear. However, it has been speculated that female hormones, particularly estrogen, may play a protective role against GIST development and growth. Additionally, female patients may have a lower incidence of comorbidities and lifestyle-related risk factors, such as smoking and heavy alcohol consumption, which are associated with worse GIST outcomes. The results obtained from our study indicated that CSS prognosis was significantly influenced by the presence of poorly and undifferentiated grade. The study conducted by Khan et al. [24] similarly revealed that a worse prognosis was associated with poorly and undifferentiated grade. Poorly and undifferentiated tumors usually indicate that the tumor cells have a greater number of abnormal features and stronger abilities in growth, division, and spreading, which may make them more difficult to treat. Additionally, these cells may invade surrounding tissues or distant organs via lymphatic or blood vessels, leading to the spread of the tumor to other parts of the body and impacting treatment efficacy and patient prognosis. Thus, the worse the differentiation of a GIST, the poorer the prognosis for patients.

Our research has some limitations. First, this is a retrospective study based on the SEER database, so data missing and bias are inevitable. However, we employed MI and PSM techniques to address missing data and selection bias. Secondly, the SEER database does not provide information on postoperative complications and recurrence, which may also affect patients' long-term prognosis. Third, while cardia GIST is rare, we were only able to identify 47 patients who underwent ET from the SEER database over the past 19 years. Although a small sample size may impact the reliability of our findings, it still represents the largest sample size of ET for cardia GIST currently available.

In conclusion, our study compared long-term survival outcomes of patients who underwent ET or SR for cardia GIST using data from the SEER database. We found that ET and SR had comparable long-term survival rates and that ET could be a valid surgical approach for treating cardia GIST. However, further exploration is needed to determine the ideal surgical strategy for this rare type of tumor.

## Author Contributions

**Conceptualization:** Zhenguo Qiao, Yimin Ma.

**Data curation:** Qiong Wu.

**Formal analysis:** Qiong Wu.

**Methodology:** Zhuofan Li.

**Resources:** Xin Ling.

**Software:** Jun Jiang.

**Supervision:** Xin Ling.

**Writing – original draft:** Qiong Wu, Jun Jiang, Zhuofan Li.

**Writing – review & editing:** Zhenguo Qiao, Yimin Ma.

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
