## [Decision Letter · Decision Letter 0]

3 Jun 2024

PONE-D-24-16551Long-term Survival Outcomes of Endoscopic Therapy vs. Surgical Resection in Patients with Cardia Gastrointestinal Stromal TumorPLOS ONE

Dear Dr. Qiao,

Thank you for submitting your manuscript to PLOS ONE. After careful consideration, we feel that it has merit but does not fully meet PLOS ONE’s publication criteria as it currently stands. Therefore, we invite you to submit a revised version of the manuscript that addresses the points raised during the review process.

We look forward to receiving your revised manuscript.

Kind regards,

Filomena de Nigris, Ph.D.

Academic Editor

PLOS ONE

Journal Requirements:

Reviewers' comments:

Reviewer's Responses to Questions

**Comments to the Author**

1. Is the manuscript technically sound, and do the data support the conclusions?

Reviewer #1: Yes

Reviewer #2: Yes

2. Has the statistical analysis been performed appropriately and rigorously? 

Reviewer #1: Yes

Reviewer #2: Yes

3. Have the authors made all data underlying the findings in their manuscript fully available?

Reviewer #1: Yes

Reviewer #2: Yes

4. Is the manuscript presented in an intelligible fashion and written in standard English?

Reviewer #1: No

Reviewer #2: Yes

5. Review Comments to the Author

Reviewer #1: The work presented is original and well structured. The statistical analysis was well documented and described in detail with supplementary tables. There is one aspect to review: authors should review the text paying attention to : punctuation, articles and maintaining the plural / singular agreement .

Reviewer #2: The study aimed to assess the long-term survival results among patients who received endoscopic therapy (ET) or surgical resection (SR) for cardia GIST. Data derived from Cardia GIST patients enrolled from 2000 to 2019 and selected from the surveillance, epidemiology, and end result (SEER) database. Multiple imputation (MI) was applied to handle missing data, and propensity score matching (PSM) was carried out to mitigate selection bias during comparisons. A total of 330 patients with cardia GIST were enrolled, including 47 (14.2%) patients with ET and 283 (85.8%) patients with SR. The 5-year OS and CSS rates in the ET and SR groups were comparable [before PSM, (OS) (76.1% vs. 81.2%, P=0.722), (CSS) (95.0% vs. 89.3%, P=0.186); after PSM, (OS) (75.4% vs. 85.4%, P=0.540), (CSS) (94.9% vs. 92.0%, P=0.099)]. Moreover, there was no significant difference between ET and SR in terms of long-term OS (hazard ratio [HR] 0.735, 95% confidence interval [CI] 0.422-1.282) and CSS (HR 1.560, 95% CI 0.543-4.481). The study is of interest. The limitations were descripted. Even the sample is small the results are of interest for clinicians.

It would be beneficial to include "No at risk" on the survival curves.

6. PLOS authors have the option to publish the peer review history of their article (what does this mean?). If published, this will include your full peer review and any attached files.

Reviewer #1: No

Reviewer #2: No

---

## [Author Response · Author response to Decision Letter 0]

12 Jun 2024

Dear editor and reviewers of PLOS ONE:

Our reference: PONE-D-24-16551

Title: Long-term Survival Outcomes of Endoscopic Therapy vs. Surgical Resection in Patients with Cardia Gastrointestinal Stromal Tumor

By: Qiong Wu et al

Thank you very much for your letter and for the editors’ and reviewers’ comments concerning our manuscript entitled "Long-term Survival Outcomes of Endoscopic Therapy vs. Surgical Resection in Patients with Cardia Gastrointestinal Stromal Tumor” (ID: PONE-D-24-16551). These comments are of great reference value to the revision and improvement of our paper and have important guiding significance to our researches. We have studied comments carefully and have made correction. We hope that the revision is acceptable and look forward to hearing from you soon. Revised portion are marked in color in the paper. The main corrections in the paper and the responds to the reviewer’s comments are as following:

Reviewer: 1

Comment 1

The work presented is original and well structured. The statistical analysis was well documented and described in detail with supplementary tables. There is one aspect to review: authors should review the text paying attention to : punctuation, articles and maintaining the plural / singular agreement .

Response 1

Thank you very much for your meticulous evaluation of our work and for providing valuable feedback. We are pleased to hear that you found our work to be original and well-structured, and our statistical analysis to be well-documented and described in detail with supplementary tables. Regarding your suggestion for text revision, we fully agree. Indeed, the accuracy and fluency of the text are crucial for conveying the research content and ensuring reader comprehension. Therefore, we will carefully review the entire text, paying particular attention to punctuation, articles, and maintaining plural/singular agreement. After a careful review, we have not found any obvious punctuation errors. We have unified all references of "gastrointestinal stromal tumors (GISTs)" to the singular form "gastrointestinal stromal tumor (GIST)".

Comment 2

The study aimed to assess the long-term survival results among patients who received endoscopic therapy (ET) or surgical resection (SR) for cardia GIST. Data derived from Cardia GIST patients enrolled from 2000 to 2019 and selected from the surveillance, epidemiology, and end result (SEER) database. Multiple imputation (MI) was applied to handle missing data, and propensity score matching (PSM) was carried out to mitigate selection bias during comparisons. A total of 330 patients with cardia GIST were enrolled, including 47 (14.2%) patients with ET and 283 (85.8%) patients with SR. The 5-year OS and CSS rates in the ET and SR groups were comparable [before PSM, (OS) (76.1% vs. 81.2%, P=0.722), (CSS) (95.0% vs. 89.3%, P=0.186); after PSM, (OS) (75.4% vs. 85.4%, P=0.540), (CSS) (94.9% vs. 92.0%, P=0.099)]. Moreover, there was no significant difference between ET and SR in terms of long-term OS (hazard ratio [HR] 0.735, 95% confidence interval [CI] 0.422-1.282) and CSS (HR 1.560, 95% CI 0.543-4.481). The study is of interest. The limitations were descripted. Even the sample is small the results are of interest for clinicians.

It would be beneficial to include "No at risk" on the survival curves.

Response 2

Thank you for your valuable feedback. We appreciate your suggestion to include the "Number at risk" on the survival curves, as it indeed provides important contextual information for interpreting the results. In response to your recommendation, we have re-analyzed our data and redrawn the survival curves using R software (version 4.0). The updated curves now feature the "Number at risk" at various time points, offering a more comprehensive understanding of the survival patterns observed in our study. In the new Figure 2, we have added "Number at risk".

---

## [Editor Report · Decision Letter 1]

21 Jun 2024

Long-term Survival Outcomes of Endoscopic Therapy vs. Surgical Resection in Patients with Cardia Gastrointestinal Stromal Tumor

PONE-D-24-16551R1

Dear Dr. Qiao

We’re pleased to inform you that your manuscript has been judged scientifically suitable for publication and will be formally accepted for publication once it meets all outstanding technical requirements.

Kind regards,

Filomena de Nigris, Ph.D.

Academic Editor

PLOS ONE

Additional Editor Comments (optional):

The rebbutal letter addressed reviewers comments so I decided to accept the manuscript for publication without any additional reviewers comments

---

## [Editor Report · Acceptance letter]

24 Jun 2024

PONE-D-24-16551R1 

PLOS ONE

Dear Dr. Qiao, 

I'm pleased to inform you that your manuscript has been deemed suitable for publication in PLOS ONE. Congratulations! Your manuscript is now being handed over to our production team.

Kind regards, 

on behalf of

Prof. Filomena de Nigris 

Academic Editor

PLOS ONE